# Single-cell transcriptomics of leukocyte-enriched human milk reveals a highly diverse and adapted myeloid compartment

Nadine Schrode[1], Xiaoqi Yang[2], Alisa Fox[2], Disha Chowhan[1], Claire DeCarlo[2], Kristin Beaumont[1], Rebecca L.R. Powell[3]*

1 Department of Genetics and Genomic Sciences, Icahn School of Medicine at Mount Sinai, New York, New York, United States of America, 2 Department of Medicine, Division of Infectious Diseases, Icahn School of Medicine at Mount Sinai, New York, New York, United States of America, 3 Department of Medicine, Division of Infectious Diseases; Department of Immunology and Immunotherapy, Icahn School of Medicine at Mount Sinai, New York, New York, United States of America

* Rebecca.Powell@mssm.edu

## Abstract

Human milk is a bioactive fluid containing maternal leukocytes that support infant immune protection and development. Prior work has demonstrated the presence of atypical, non-granulocyte milk myeloid cells that do not conform to previously-defined cell subsets. To assess the true diversity of mononuclear milk myeloid cells, we performed single-cell RNA sequencing (scRNAseq) on >23,000 CD45+ enriched cells from 9 healthy lactating donors. Clustering identified 18 transcriptionally distinct mononuclear myeloid subpopulations, including 10 macrophage clusters, 5 dendritic cell subsets, 1 monocyte population, and 2 epithelial-like clusters. Macrophages segregated into M1-like and M2-like states exhibiting discrete transcriptional programs. The epithelial-like clusters expressed both epithelial markers (e.g., EPCAM, KRT19, CLDN3) and myeloid genes, suggesting phagocytic activity or hybrid states. Numerous myeloid sub-clusters exhibited transcriptional features consistent with adaptation to the lipid-rich environment of milk, supported also by reactome data. This included LDL and lipoprotein clearance pathways and expression of genes such as INSIG1 and QSOX1 in monocytes and APOE, LIPA, and APOC1 in macrophages. Signaling pathways were also dominated by degranulation and IL-4/10/13 activities, as well as antigen presentation programs. CellChat analysis revealed extensive intercellular communication among myeloid subsets involving MHC-II, SPP1, and MIF, consistent with active antigen presentation and tissue remodeling. Notably, pronounced donor-specific heterogeneity was observed, with several sub-clusters restricted to 1–2 individuals, underscoring personalized immune composition during lactation. These data indicate that milk myeloid cells are not merely passively transported immune cells but actively shaped and diversified by the lactational niche, and establish a high-resolution framework for future studies.

**Data availability statement:** All relevant data are within the manuscript and its Supporting Information files.

**Funding:** This study was funded by a grant from The Campbell Foundation. The funders had no role in study design, data collection and analysis, decision to publish, or preparation of the manuscript.

**Competing interests:** The authors have declared that no competing interests exist.

## Introduction

Human milk is a dynamic fluid that extends beyond nutrition to provide vital immunological protection and developmental cues to the infant. Human milk-fed infants ingest ~$10^5$–$10^8$ maternal leukocytes daily, though very few studies have extensively identified the leukocyte subsets particular to human milk [1]. Animal studies have demonstrated that human milk leukocytes can survive gastrointestinal (GI) transit and migrate into neonatal lymphoid tissues, contributing to early immune programming and protection. In a recent study, milk-derived T cells were shown to enhance responses to infection and vaccination [2]. Earlier rodent studies similarly found that milk macrophages and lymphocytes localize to Peyer's patches and act as immune modulators [3–5]. The long-term development and programming of the infant immune system by maternal milk leukocytes are believed to contribute to reduced risks of allergic and autoimmune diseases later in life, establishing human milk as a key determinant of early-life immune imprinting [6,7]. A more refined understanding of the cellular subsets present in milk may help explain how maternal immune factors influence systemic and mucosal immune development in the infant as well as protection from infection. As well, if comprehensively studied, milk may offer a non-invasive window into assessing maternal mucosal immune status, with changes in milk cell composition reflecting systemic inflammation, infection, or autoimmune activity. Profiling these cells at high resolution may yield novel biomarkers of postpartum health and may inform precision approaches to donor milk selection, fortification, or therapeutic supplementation for at-risk neonates.

Prior work by Trend et al. [8] utilized flow cytometry to profile leukocyte populations in human colostrum and mature milk, revealing that macrophages constituted the dominant immune cell type, accompanied by monocytes, neutrophils, and lymphocytes. Intriguingly, they identified a population of mononuclear (non-granulocyte) myeloid cells that lacked classical maturation markers or expressed them in non-canonical combinations. This suggested the presence of immature or transitional myeloid subsets. Our recent studies demonstrated that various milk myeloid cell types can perform antibody dependent cellular phagocytosis (ADCP) [9]. Using PBMC gating strategies, this work identified classical monocytes, activated monocytes/macrophages, granulocytes, and dendritic cells [9]. Additionally, we identified similar to prior data a significant population of atypical, non-granulocyte $CD14^-$ milk myeloid cells that did not conform to any defined cell subset, many of which appeared capable of ADCP. These findings raised important questions about the ontogeny and functional roles of these atypical cells, including whether they represented developmental intermediates, cells actively differentiating within the milk microenvironment, or a unique myeloid phenotype induced by local cues.

Despite the central role of immune cells in milk, relatively few studies have leveraged high-resolution techniques to fully characterize their diversity. Single-cell RNA sequencing (scRNAseq) is a powerful method that is used to evaluate differences in gene expression among individual subpopulations of cells. This approach elucidates heterogeneity in complex biological systems and allows for the investigation of low frequency cellular compartments, which are often overlooked in bulk RNA

sequencing. Recently, Nyquist et al. [10] applied scRNAseq to human milk samples and revealed a spectrum of epithelial and immune cells across the period of lactation. However, their study did not enrich for immune populations prior to sequencing, resulting in a dataset heavily skewed toward epithelial transcripts and offering limited resolution of the myeloid compartment.

In the present study, we applied scRNAseq on sorted CD45 + human milk cells, with a specific focus on classifying the atypical myeloid compartment. By enriching for immune cells prior to sequencing, we aimed to overcome the limitations of prior studies and generate a high-resolution map of macrophage, monocyte, and dendritic cell populations in milk.

## Results

In the present analysis, ~ 100 - 350mL fresh milk was collected from 9 unique donors at various stages of mature lactation (Table 1). After initial gating out of debris, doublets and dead cells, CD45 + DRAQ5 + cells were sorted for scRNAseq analysis. Milk samples were found to contain 0.3–27% CD45 + cells, consistent with prior published data [9]. After passing QC requirements, a total of 23443 cells were analyzed.

### Annotation of broad cell types

Canonical marker gene expression was used to define major cell types (Fig 1) [11]. Cells expressing CD79A and MS4A1 were annotated as B cells. These cells also expressed low levels of CD4. Cells expressing CD3D in addition to co-expression of CD8A and/or CD4 were annotated as T cells. Cells expressing CD8A without CD4 were classified as CD8 T cells, while cells expressing CD4 were classified as CD4 T cells. Cells exhibiting high expression of CST3 were annotated as dendritic cells. Co-expression of CD68 and CD14 was also observed. Cells exhibiting co-expression of CD68, CD163, and CD14 were annotated as macrophages. This population also exhibited high expression of the CST3 marker, low expression of CD4, and moderate expression of the epithelial marker PHLDA1. Epithelial cells were defined based on expression of EPCAM with co-expression of CDH1, MUC1, ELF3, CLDN3, CLDN4, and PHLDA1 at varying levels for the glandular, ductal, luminal, and endo-epithelial subsets.

### Myeloid cell annotation

The top 10 differentially expressed genes (DEGs) for each major myeloid cell type (based on average log-fold change and statistical significance using a threshold of log-fold change > 0.7 and adjusted p < 0.05) representing typically uncharacterized components of each type's transcriptomic identity are described here. Full DEG data is available in Supplemental Data 1, though the focus of this study was mononuclear myeloid cells, including monocytes, macrophages, and dendritic

**Table 1. Sample data.**

| | Age | Race/Ethnicity | Months Postpartum | Sample Volume (mL) | Total cells | Cells/mL | CD45+(% gated) | Cell count after FACS/filter | Cell count after QC |
|---|---|---|---|---|---|---|---|---|---|
| S201 | 44 | White | 2.7 | 100 | 7.2E + 06 | 7.2E + 04 | 4% | 1881 | 1427 |
| S202 | 36 | White | 4.6 | 100 | 3.8E + 06 | 3.8E + 04 | 8% | 2378 | 1914 |
| S204C | 38 | White | 15.7 | 110 | 1.40E + 07 | 1.3E + 05 | 0.9% | 658 | 557 |
| S205 | 31 | White/Hispanic | 10.8 | 170 | 6.00E + 06 | 3.5E + 04 | 2.6% | 30704 | 12526 |
| S206 | 33 | White | 7.3 | 170 | 6.50E + 06 | 3.8E + 04 | 5.2% | 1432 | 861 |
| S207 | 32 | White | 13.8 | 180 | 1.40E + 07 | 7.8E + 04 | 2.4% | 2156 | 1421 |
| S208 | NA | White | 9.2 | 110 | 3.50E + 06 | 3.2E + 04 | 5.7% | 469 | 276 |
| S209 | 38 | Asian | 1.8 | 350 | 8.70E + 06 | 2.5E + 04 | 27% | 2576 | 1357 |
| S210 | 35 | White | 9.8 | 150 | 2.48E + 07 | 1.7E + 05 | 4.9% | 13258 | 3116 |

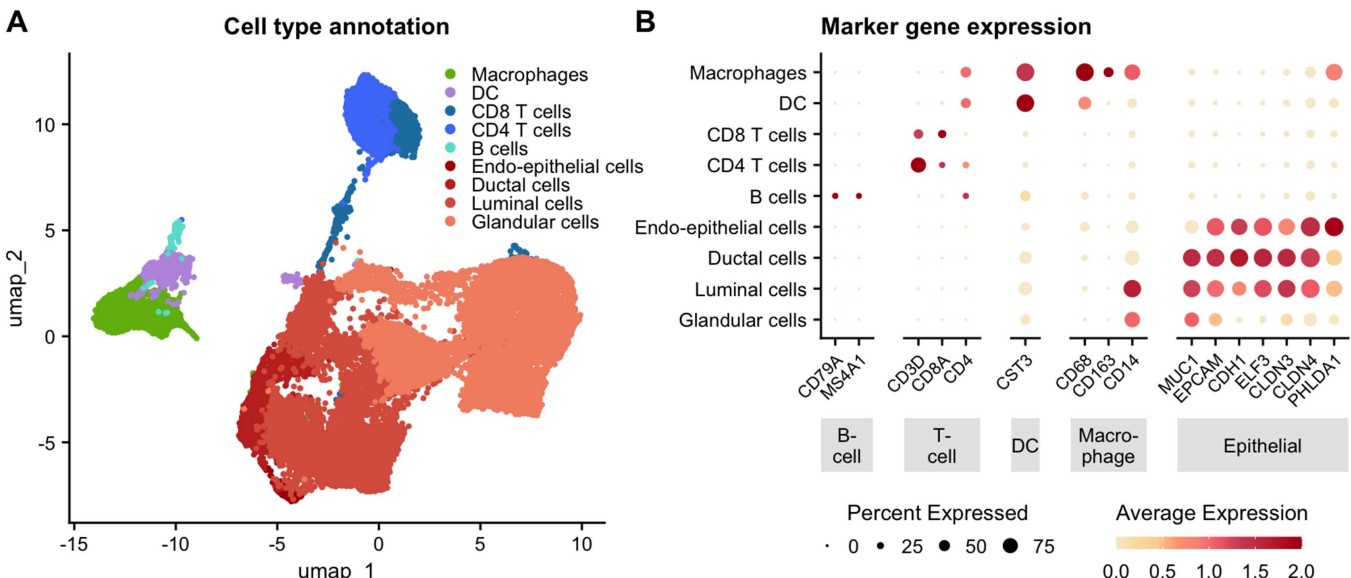

**Fig 1. Single-cell RNA sequencing reveals major cell types in human milk after leukocyte enrichment. (A)** UMAP projection of 23,443 CD45+DRAQ5+ cells isolated from fresh milk samples (100-350 mL) collected from 9 healthy lactating donors at various stages of mature lactation. Cells were sorted following exclusion of debris, doublets, and dead cells, and processed using the 10x Genomics Chromium 3′ GEX V3 platform. Libraries were sequenced on a NovaSeq to a target depth of ≥50,000 reads/cell. Post-alignment with Cell Ranger (v7.1), data were processed using Seurat (v3.1.1) with SCTransform normalization, CCA integration, PCA dimensionality reduction, and graph-based Louvain clustering. **(B)** Dot plot of canonical lineage marker genes used for cell type annotation. Dot size reflects the percentage of cells expressing each gene within a cluster, and color intensity indicates the average scaled expression.

cells, as described below (Fig 2). Note that apoptotic myeloid cell subsets (C8_0, C8_1) were also identified, by a highly increased mitochondrial gene expression ratio and enrichment for genes associated with apoptotic cleavage (Fig 2). DEGs were evaluated against curated reference datasets, including the Gene Expression Omnibus and Human Protein Atlas and considered non-canonical/atypical if they showed minimal prior association with myeloid lineages, were predominantly linked to non-myeloid cell types, or represented context-restricted expression not previously described in immune cells.

## Monocytes

Several of the top monocyte-enriched DEGs showed limited prior characterization in myeloid cells, suggesting potential context-specific expression within the lactating mammary gland. Lipid handling and secretory stress adaptation may be supported by INSIG1, a regulator of cholesterol biosynthesis, and SLC7A5, an amino acid transporter that activates mTOR signaling; both are broadly associated with metabolic regulation and may reflect adaptation to the lipid-rich environment of milk. PPIF, which regulates mitochondrial permeability and oxidative stress, and NIBAN2, a gene involved in ER stress signaling, may support cellular resilience under secretory pressure, though their roles in monocytes remain largely unexplored. Stromal remodeling and ductal migration may be influenced by DSE, an ECM-modifying enzyme that alters glycosaminoglycan composition, and QSOX1, which promotes matrix adhesion through disulfide bond formation, functions that could be relevant for monocyte trafficking in a densely glandular tissue. Motility and cytoskeletal remodeling included TPM4 and EFHD2, regulators of actin filament dynamics and calcium-responsive cytoskeletal rearrangement, respectively. Finally, transcriptomic plasticity and vascular-immune interaction could be inferred by expression of HNRNPM, a

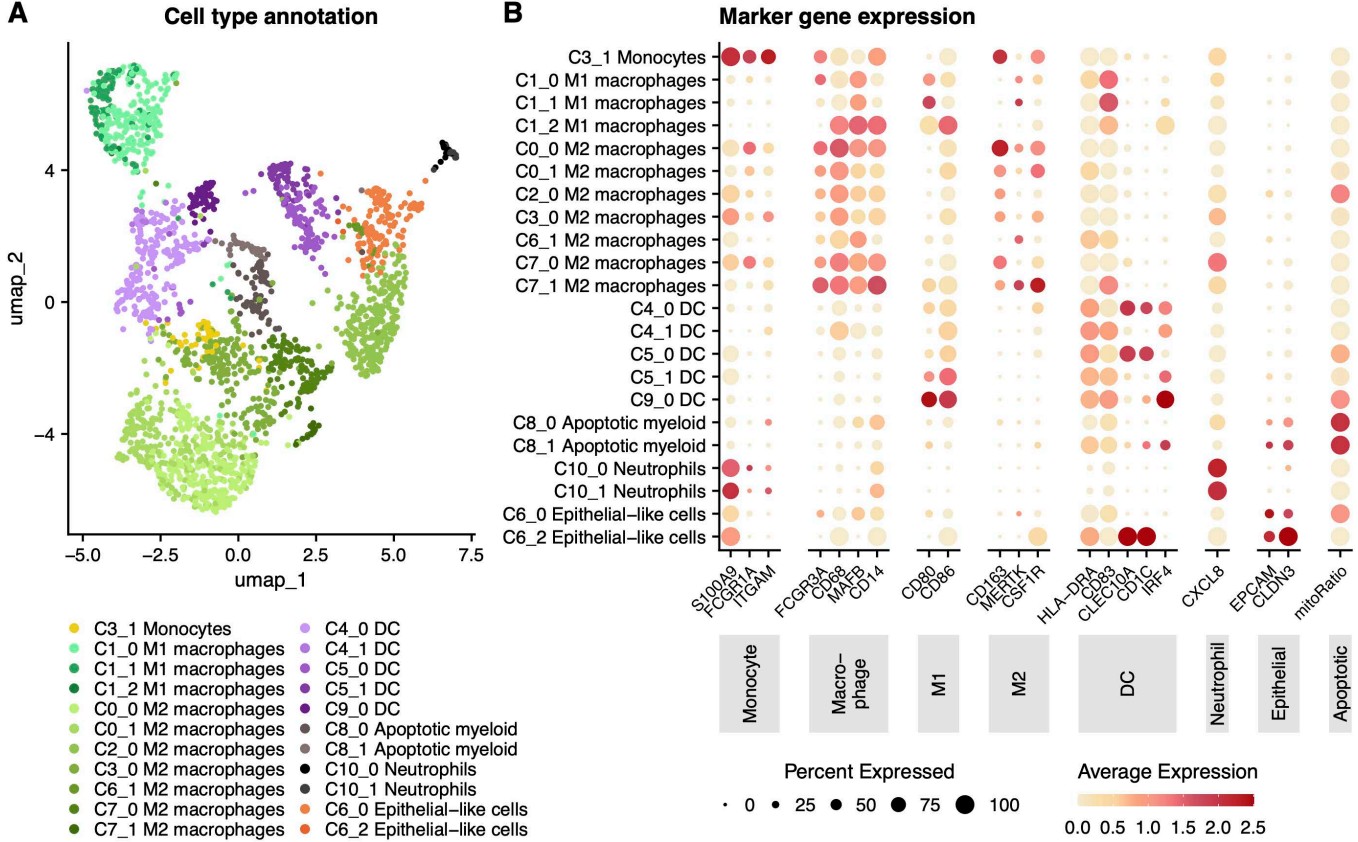

**Fig 2. High-resolution sub-clustering of myeloid cells in human milk. (A)** UMAP projection of cells classified within the myeloid compartment, reanalyzed as in Fig 1 using PCA, UMAP, and Louvain clustering. Sub-clusters were assigned based on canonical marker expression and validated using supervised classifiers trained on public datasets and deep-learning deconvolution via Unicell. Epithelial-like subpopulations were retained based on dual expression of immune and epithelial markers. **(B)** Dot plot showing expression of representative genes across identified sub-clusters. Dot size indicates the proportion of expressing cells per cluster; color indicates mean scaled expression.

splicing factor linked to myeloid differentiation, and ECE1, an endothelin-converting enzyme that could mediate communication between monocytes and local vasculature.

**M1 Macrophages**

Among DEGs enriched in the M1 macrophage population, several genes were identified with little prior association to myeloid cells, raising the possibility of a milk-adapted inflammatory phenotype. NPTX2, a neuronal pentraxin involved in complement regulation and synaptic pruning, may point to cross-talk with neuroimmune pathways or atypical complement handling. Similarly, IGSF21, an adhesion molecule with synaptogenic roles in inhibitory neurons, may reflect novel interactions with epithelial or neural elements in the tissue niche. A number of cytoskeletal and transport-related genes with developmental or structural specificity also emerge, including XIRP1, known for actin-binding in muscle and epithelial cells; KIF26B, a non-canonical kinesin with roles in morphogenesis and cytoskeletal remodeling; and KIF1A, a neuronal motor protein essential for axonal vesicle transport, here possibly supporting macrophage trafficking or intracellular vesicle dynamics. CNN1 (calponin), a smooth muscle contractile protein, may suggest matrix-sensing or structural remodeling properties in activated macrophages. Ion and signaling-associated genes such as SYT6, a synaptotagmin isoform

functioning as a calcium sensor, and SLC16A2, a thyroid hormone transporter, raise the possibility of novel vesicular or metabolic responsiveness. ELAVL4, a neuronal RNA-binding protein, and ADGRB3, an adhesion GPCR involved in synaptic pruning, suggest the emergence of noncanonical transcriptional and adhesive programs within macrophages responding to complex microenvironmental signals.

## M2 Macrophages

Several DEGs enriched in the M2-like macrophage cluster exhibited limited or no prior characterization in monocytes or macrophages, suggesting potential adaptation to the lactating mammary microenvironment. Neuroendocrine and epithelial crosstalk candidates included AGRP, a neuropeptide typically restricted to the hypothalamus, and CELSR1, a planar polarity gene associated with epithelial organization; although not traditionally linked to immune cells, their expression may reflect transcriptional influences from neighboring neuroepithelial or ductal tissues in milk. Extracellular matrix and structural remodeling was represented by MFAP5, an ECM glycoprotein with sparse immune annotation, and HS3ST2, a sulfotransferase involved in heparan sulfate biosynthesis; both may contribute to stromal remodeling or tissue-specific interactions encountered by macrophages in glandular tissue. Metabolic specialization and oxidative adaptation included ACOX2, an enzyme involved in peroxisomal β-oxidation and bile acid metabolism, JAKMIP2, a vesicle trafficking protein primarily studied in neurons, and SLC38A6, a sodium-dependent amino acid transporter of unclear immune relevance; these may reflect alternative nutrient handling pathways adapted to the milk environment. Stress response and differentiation regulators such as VWA5A and MAP3K7CL, both poorly annotated in immune contexts, may reflect transcriptional plasticity or tissue-specific adaptation. TM4SF19, a tetraspanin-like membrane protein of unknown function, further highlights the potential for uncharacterized signaling programs in this milk-resident macrophage population.

## Dendritic Cells

Several DEGs enriched in the dendritic cell cluster showed minimal or no prior association with dendritic or myeloid cell biology, suggesting the possibility of milk-specific transcriptional programs in the lactating mammary environment. CD5, a lymphocyte-associated co-receptor, and GCSAM, a germinal center protein with B cell specificity, suggest potential cross-lineage transcriptional activity or atypical immune signaling roles. Two G-protein-coupled receptors, P2RY10 and GPR171, typically linked to lysophospholipid and neuropeptide signaling respectively, may reflect novel sensory or metabolic interfaces in tissue-resident DCs. PTGDS, a prostaglandin D2 synthase largely characterized in epithelial and neural tissues, implies a capacity for local lipid mediator production distinct from canonical myeloid eicosanoid pathways. ENHO (adropin), a metabolic peptide hormone, may indicate energetic coupling to epithelial or systemic metabolic states. The presence of C17orf99 (IL-40), a recently described cytokine associated with humoral immunity, suggests unconventional communication axes within the mammary or mucosal immune environment. Genes such as SLC4A3, a bicarbonate transporter, and DNAH11, a component of motile cilia machinery, are largely unstudied in immune contexts but may hint at structural or ion-handling specializations in duct-associated DCs. Finally, PPP1R14A, an inhibitor of myosin phosphatase, may support cytoskeletal regulation or migratory dynamics unique to the stromal or ductal niche.

## Epithelial-like Cells

Cell polarity and vesicular trafficking programs included CRB3, a polarity determinant involved in tight junction formation, and MAL2, a mediator of apical protein sorting, both of which may support membrane organization or reflect epithelial imprinting. SYNE4, linking the nuclear envelope to the cytoskeleton, and ARHGEF35, a Rho- Rho guanine nucleotide exchange factor, may contribute to cytoskeletal tension and compartmentalization in tissue-adapted cells. Hormonal and developmental signaling was represented by LIFR, a key receptor in mammary morphogenesis and regeneration, and NR2F2, a nuclear receptor that regulates differentiation and paracrine signaling in glandular tissues. ADGRF1, an

adhesion G protein-coupled receptor, and GABRE, a GABA receptor subunit with non-neuronal expression in glandular niches, may enable environmental sensing or local immune-epithelial communication. Finally, mucosal tolerance and structural homeostasis were suggested by STATH, a secreted protein with antimicrobial properties in saliva and milk, and PSG4, a pregnancy-specific glycoprotein known to promote regulatory cytokines such as IL-10 and TGF-β.

### Myeloid sub-cluster annotation

To resolve the significant transcriptional heterogeneity identified in the myeloid compartment of these milk immune cell-enriched samples, differential gene expression analysis was performed using 20 canonical genes (Fig 2), This allowed for the annotation of 19 transcriptionally distinct mononuclear myeloid subpopulations.

### Individual Sample Contribution across Myeloid Sub-Clusters

This analysis revealed significant sample-to-sample variation in cell type composition, as evidenced by the highly uneven distribution of donor contributions across sub-clusters (Fig 3). In the monocyte cluster C3_1, the majority of cells originated from donor S208 (50.9%) and S207 (24.6%), with smaller contributions from S202 (8.8%), S210 (8.8%), S201 (3.5%), S204C (1.8%), and S209 (1.8%). M1 macrophage sub-clusters were strongly donor-skewed: C1_0 was dominated by S207 (59.8%) and S209 (19.6%), with additional inputs from S210 (6.4%), S206 (9.8%), S201 (2.5%), S202 (1.0%), and S204C (1.0%). C1_1 consisted predominantly of cells from S207 (92.8%), with minor contributions from S206 (3.1%) and S209 (4.1%), while C1_2 was entirely derived from S207. M2 macrophage clusters exhibited both extremes: C0_0 was largely composed of cells from S202 (70.1%), followed by S201 (8.8%), S208 (4.7%), S207 (5.5%), S210 (2.2%), S204C (1.5%), S205 (0.4%), and S209 (0.7%), and C0_1 similarly had S202 as the dominant contributor (85.5%), with lesser input from S206 (5.5%), S201 (3.8%), S207 (3.8%), S208 (1.7%), S210 (1.3%), S209 (0.9%), and S204C (0.4%). C2_0 was almost entirely composed of cells from S205 (98.5%), with small contributions from S209 (1.1%) and S207 (0.4%). C3_0 was formed primarily from S207 (56.5%), S210 (20.4%), and S208 (16.7%), with minor contributions from S202 (2.2%), S204C (1.1%), S205 (0.5%), S206 (0.5%), S201 (0.5%), and S209 (1.6%). C6_1 was nearly exclusive to S205 (93.3%), with the remainder from S210 (6.7%). C7_0 showed strong skew toward S209 (85.2%), with additional

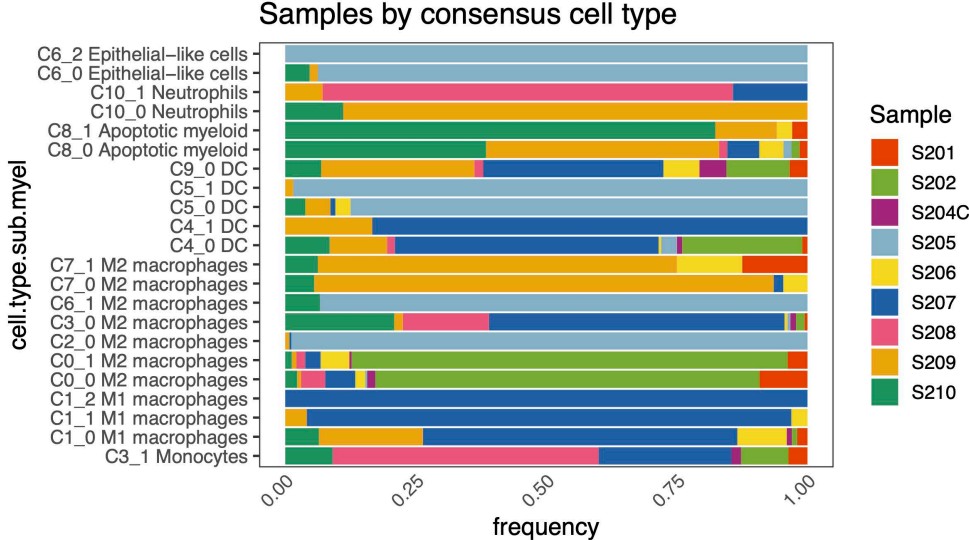

**Fig 3. Inter-individual variation in myeloid sub-cluster composition across human milk donors.** Stacked bar plot shows the proportional distribution of cells from each donor sample across all myeloid sub-clusters.

input from S210 (6.1%), S206 (5.2%), and S207 (1.7%), while C7_1 was led by S209 (64.7%), followed by S201 (11.8%), S206 (11.8%), S210 (5.9%), and S207 (5.9%). Dendritic cell sub-clusters displayed varying levels of heterogeneity. C9_0 had a more distributed composition, with cells from S207 (34.5%), S209 (29.3%), S202 (12.1%), S206 (6.9%), S204C (5.2%), S210 (6.9%), S201 (3.4%), and S208 (1.7%). C4_0 included cells from S207 (51.2%), S202 (23.7%), S209 (10.2%), S210 (8.4%), S205 (2.8%), S204C (0.9%), S201 (0.9%), and S208 (1.4%), whereas C4_1 was highly skewed toward S207 (83.3%) and S209 (16.7%). C5_0 was dominated by S205 (86.4%), with smaller fractions from S209 (5.5%), S210 (3.6%), S206 (2.7%), S207 (0.9%), and S204C (0.9%). C5_1 was even more restricted, with 97.1% of cells from S205, and 1.5% each from S207 and S209. Of the apoptotic myeloid cells, C8_0 was composed mainly of cells from S209 (34.1%) and S210 (29.4%), with additional input from S207 (4.7%), S206 (3.5%), and smaller contributions from S208, S202, S201, and S205 (each ~1%). C8_1 was dominated by S210 (66.7%), followed by S209 (9.5%), and small fractions from S201, S206, and others. Epithelial-like clusters also showed exclusive patterns, with C6_0 derived almost entirely from S205 (92.4%), along with S210 (4.6%) and S209 (3.1%), while C6_2 was restricted entirely to S205.

## Myeloid Sub-Cluster Differential Expression Analysis

The top 15 DEGs are summarized in Table 2 and Fig 4. See Supplemental Data 1 for full DEG lists and detailed description of Table 2 myeloid sub-cluster DEGs.

## Myeloid Reactome Enrichment analysis

The top 5 myeloid sub-cluster reactome pathways by z-score are summarized in Table 3. See Supplemental Data 2 for full reactome data and detailed descriptions. Additionally, to identify Reactome pathways that were both strongly enriched and broadly represented across the myeloid compartment, we calculated a weighted z-score by multiplying the total additive z-score (restricted to positive contributions) by the number of contributing sub-clusters (Fig 5). The cytokine and chemokine signaling group contained the highest-scoring pathways. *Neutrophil degranulation* (total z = 209.9; weighted z = 3148.7; 15 sub-clusters) ranked first overall, followed by *Interleukin-10 signaling* (166.3; 2494.8; 15 sub-clusters), *Signaling by Interleukins* (119.4; 1910.3; 16 sub-clusters), *Interleukin-4 and Interleukin-13 signaling* (121.3; 1818.9; 15 sub-clusters), and *Chemokine receptors bind chemokines* (117.9; 1650.6; 14 sub-clusters). These pathways were primarily driven by M2 macrophages (32–45%) and DCs (30–45%), representing the dominant cytokine-mediated signaling axis in the myeloid compartment. Following these cytokine modules, several platelet-associated and vesicular signaling pathways were highly enriched. These included *Response to elevated platelet cytosolic Ca²⁺* (total z = 81.9; weighted z = 1229.3; 15 sub-clusters), *Platelet degranulation* (81.1; 1215.9; 15 sub-clusters), and *Platelet activation, signaling and aggregation* (75.4; 1131.0; 15 sub-clusters). The major contributors were M2 macrophages (37–38%) and DCs (24–25%). Although annotated as platelet pathways in Reactome, the underlying molecular components represent calcium-dependent vesicle trafficking, integrin activation, and exocytic machinery broadly expressed in immune cells. The antigen presentation and innate immune recognition category included *Antigen processing-Cross presentation* (total z = 86.7; weighted z = 1213.4; 14 sub-clusters), *MHC class II antigen presentation* (74.6; 1044.5; 14 sub-clusters), *CLEC7A (Dectin-1) signaling* (56.9; 853.8; 15 sub-clusters), and *Toll-like Receptor Cascades* (60.5; 846.8; 14 sub-clusters). These pathways involved DCs (20–37%), M2 macrophages (33–48%), and M1 macrophages (22–25%), reflecting broad activation of antigen-loading and pathogen-sensing programs across myeloid subsets. A distinct group of lipid metabolism and clearance pathways ranked among the top 20, including *LDL clearance* (total z = 76.4; weighted z = 1069.8; 14 sub-clusters), *Plasma lipoprotein clearance* (75.2; 1052.8; 14 sub-clusters), and *Plasma lipoprotein assembly, remodeling, and clearance* (61.8; 865.5; 14 sub-clusters). These modules were predominantly driven by M2 macrophages (27–34%) and DCs (30–35%), consistent with broad representation of lipid transport and remodeling pathways across macrophage subsets.

**Table 2. Key Myeloid Sub-cluster DEGs\*.**

| Cell Type/Cluster | Key Differentially Expressed Genes | Functional Highlights |
|---|---|---|
| M1 Macrophages/C1_0 | NMRK2, OLFML3, MT3, NUAK1, HIST3H2BB, SHC3, SEMA6D, FCGBP, GLDN | Cellular metabolism, mitochondrial function, redox response, chromatin remodeling, mucosal immunity |
| M1 Macrophages/C1_1 | TWIST1, WNT11, MEG3, CTTNBP2, NPTX2, XIRP1, PDCD1, PDPN, PLVAP, NMRK2, HECTD2, ATP8B2 | Cytoskeletal dynamics, transcriptional reprogramming, immune checkpoint control, synaptic signaling, metabolic adaptation |
| M2 Macrophages/C0_0 | SIGLEC1, CCL7 | Immunomodulation, chemotaxis, innate-adaptive immune crosstalk |
| M2 Macrophages/C0_1 | THBS1, ACKR3, COLGALT2, RGMA, PLCB1 | Matrix remodeling, chemokine responsiveness, neuro-immune signaling, calcium flux regulation |
| M2 Macrophages/C2_0 | IFNB1, IFIT2, IFIT3, DNAAF1, TNIP3 | Type I interferon signaling, antiviral response, immunoregulatory profile |
| M2 Macrophages/C3_0 | S100A12, MCEMP1, FCAR, TKTL1, SIGLEC5, SERPINB2, HEY1, LILRA2, ADGRE3, JAKMIP2, TGM2, OXER1, HPSE | Inflammatory signaling, mucosal immunity, metabolic remodeling, extracellular matrix interaction |
| M2 Macrophages/C6_1 | TPSB2, H1F0, HIST2H2BE, FAM110B, FGFR1 | Mast cell phenotype, chromatin regulation, stromal remodeling, epigenetic reprogramming |
| M2 Macrophages/C7_0 | ACOD1 (IRG1) | Metabolic immune modulation, inflammasome regulation |
| M2 Macrophages/C7_1 | MMP14, SLCO2B1, IL20RB, C1QC | Tissue remodeling, complement activation, metabolic responsiveness, immune resolution |
| Dendritic Cells/C4_0 | CD207 (Langerin), CD1A, P2RY14, PTGDS | Skin/mucosal antigen presentation, glycolipid antigen processing, prostaglandin modulation |
| Dendritic Cells/C4_1 | GPR31 | Lipid signaling, transepithelial dendrite formation, inflammatory endothelial dysfunction |
| Dendritic Cells/C5_0 | CD1E, CD1C, FCER1A, LGALS2 | Lipid antigen presentation, allergen uptake, immune modulation |
| Dendritic Cells/C5_1 | PODXL2, CCR6, FLT3, LAD1 | Immune cell trafficking, barrier surveillance, dendritic cell development |
| Dendritic Cells/C9_0 | CD200, DNAH11, TTLL9, GNG11, TNNT2 | Immune regulation, cytoskeletal remodeling, vesicular trafficking, inflammatory signaling |
| Epithelial-like Myeloid Cells/C6_0 | MAP1B, ADGRF1, NR2F2, LIFR | Cytoskeletal organization, extracellular matrix interaction, macrophage polarization, tissue repair |

*Top 15 differentially expressed genes (DEGs) for each cell cluster based on average log-fold change with statistical significance using a threshold of log-fold change > 0.7 and adjusted p < 0.05. Sub-clusters with up to DEGs reaching this threshold are listed.*

Finally, the metabolic stress and inflammatory response group included *Purinergic signaling in leishmaniasis infection* (total z = 86.1; weighted z = 1118.7; 13 sub-clusters), *Cell recruitment (pro-inflammatory response)* (86.1; 1118.7; 13 sub-clusters), *ROS and RNS production in phagocytes* (73.6; weighted z = 957.0; 13 sub-clusters), *Cellular response to starvation* (89.3; weighted z = 893.2; 10 sub-clusters), and *Interferon gamma signaling* (67.7; weighted z = 879.6; 13 sub-clusters). These pathways were dominated by DCs (37–68%), M2 macrophages (35–45%), and M1 macrophages (25%).

To further investigate the lipid metabolism and clearance pathways evident from the DEG and Reactome data, we performed co-localization analysis of canonical monocytes and macrophage markers (S100A9, FCGR1A, ITGAM and FCGR3A, CD68, MAFB, CD14, respectively, with various lipid-handling genes (INSIG1, SLC7A5, QSOX1 for monocytes or APOE, LIPA, APOC1, NCEH1, SOAT1, A2M for macrophages). This analysis verified distinct subsets of monocytes and macrophages exhibiting significant lipid-handling transcriptomic signatures (Fig 6).

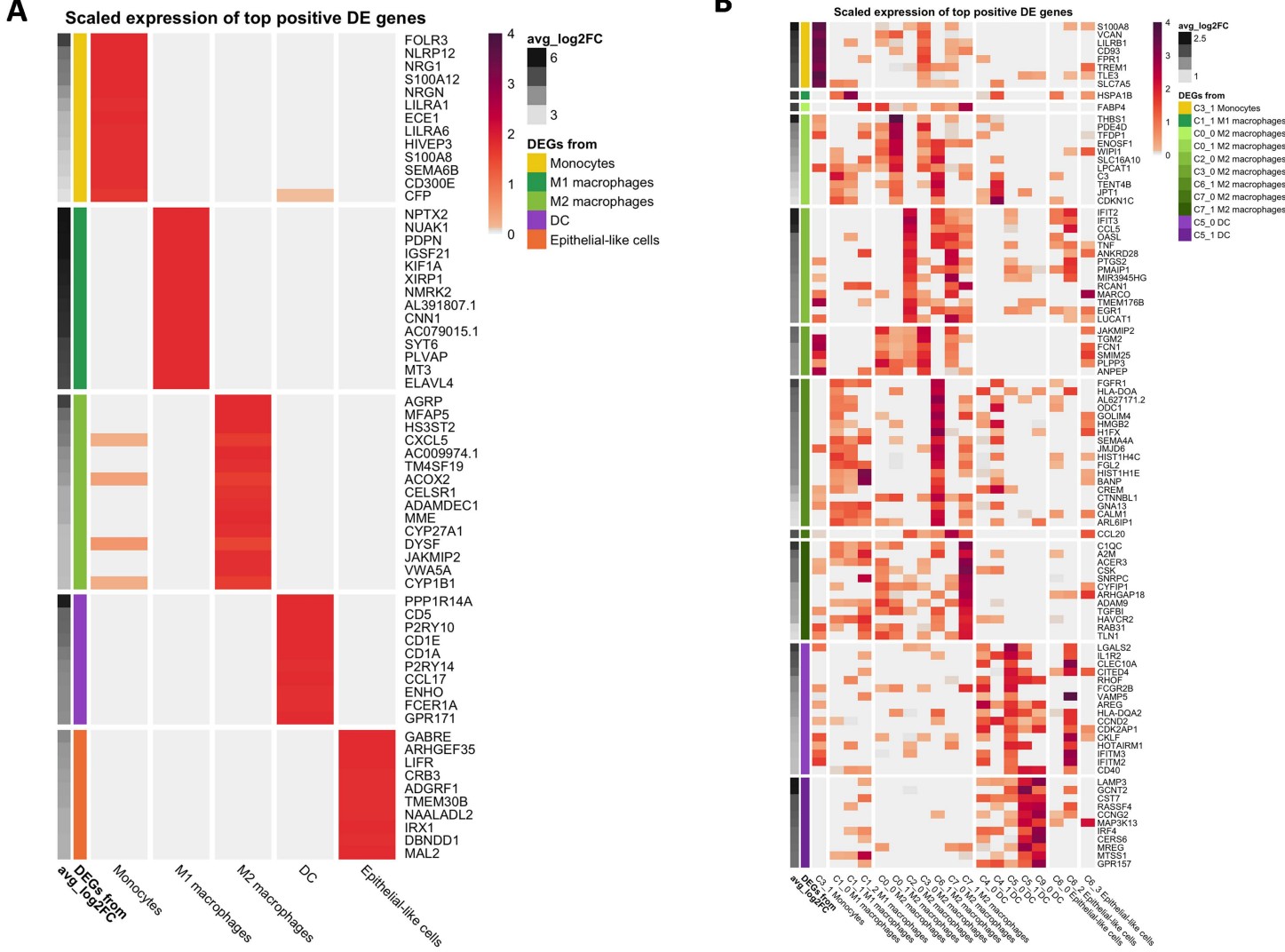

**Fig 4. Scaled expression of top DEGs across broad mononuclear myeloid cell types and sub-clusters. (a)** Broad myeloid cell type DEGs. **(b)** Sub-cluster DEGs. Analysis was performed using the FindMarkers function in Seurat. Genes were considered significantly upregulated in each cluster if positive log2 fold-change (avg_log2FC>0) and adjusted p-value<0.05. The top positive DEGs for each sub-cluster were selected and their expression scaled (z-score normalization across all clusters) to visualize relative expression levels. Columns represent annotated sub-clusters; rows represent top DEGs enriched in each cluster. Color intensity corresponds to scaled average expression; genes with higher relative expression in a given sub-cluster appear darker. Sub-cluster of DEG origin is denoted by the color-coded bar at left.

## Inferred Myeloid Cell-Cell communication networks

Complete CellChat analysis is available in supplemental data 3. The 5 most active signaling pathways, as defined by total signaling score, included MHC-II, SPP1 (osteopontin), MIF (macrophage migration inhibitory factor), MHC-I, and CCL chemokines.

Restricting the analysis to signals originating from myeloid cells, the MHC-II signaling pathway emerged as the most prominent, with a total signaling score of 86.62, an average interaction strength of 0.45, and 191 unique interactions. This pathway was actively sent by multiple myeloid subsets, including M2 macrophages (C0_0, C0_1, C2_0, C3_0, C6_1,

**Table 3. Top Myeloid Sub-cluster Reactome pathways.**

| Myeloid Sub-cluster | Top Reactome Pathways by z-score | Putative Role(s) in Mammary/Milk Environment |
|---|---|---|
| **C3_1 Monocytes** | Plasma lipoprotein clearance<br>LDL clearance<br>Plasma lipoprotein assembly, remodeling, and clearance<br>Neutrophil degranulation<br>Cytokine signaling in immune system | Lipid transport and metabolism<br>Immune effector function |
| **C1_0 M1 Macrophages** | Respiratory electron transport, ATP synthesis by chemiosmotic coupling, and heat production by uncoupling proteins<br>The citric acid (TCA) cycle and respiratory electron transport<br>Respiratory electron transport<br>Formation of ATP by chemiosmotic coupling<br>Neutrophil degranulation | Mitochondrial metabolism<br>Immune effector function |
| **C1_1 M1 Macrophages** | Formation of a pool of free 40S subunits<br>GTP hydrolysis and joining of the 60S ribosomal subunit<br>SRP-dependent co-translational protein targeting to membrane<br>L13a-mediated translational silencing of Ceruloplasmin expression<br>Attenuation phase | Protein synthesis and translational control |
| **C0_0 M2 Macrophages** | Respiratory electron transport, ATP synthesis by chemiosmotic coupling, and heat production by uncoupling proteins<br>Respiratory electron transport<br>The citric acid (TCA) cycle and respiratory electron transport<br>Complex I biogenesis<br>Neutrophil degranulation | Mitochondrial metabolism<br>Immune effector function |
| **C0_1 M2 Macrophages** | Interleukin-10 signaling<br>Chemokine receptors bind chemokines<br>LDL clearance<br>Aberrant regulation of mitotic G1/S transition in cancer due to RB1 defects<br>Defective binding of RB1 mutants to E2F1,(E2F2, E2F3) | Immunoregulation<br>Lipid handling<br>Cell cycle control |
| **C2_0 M2 Macrophages** | Interleukin-10 signaling<br>Chemokine receptors bind chemokines<br>Neutrophil degranulation<br>Respiratory electron transport<br>Complex I biogenesis | Immunoregulation<br>Mitochondrial metabolism<br>Immune effector function |
| **C3_0 M2 Macrophages** | Neutrophil degranulation<br>Interleukin-10 signaling<br>PD-1 signaling<br>Translocation of ZAP-70 to Immunological synapse<br>Phosphorylation of CD3 and TCR zeta chains | Immune effector function<br>T cell interaction<br>Immunoregulation |
| **C6_1 M2 Macrophages** | Other semaphorin interactions<br>Signal transduction by L1<br>Tetrahydrobiopterin (BH4) synthesis, recycling, salvage and regulation<br>MET receptor recycling<br>Ionotropic activity of kainate receptors | Redox regulation<br>Membrane protein recycling<br>Cell-cell signaling |
| **C7_0 M2 Macrophages** | Neutrophil degranulation<br>Translocation of ZAP-70 to Immunological synapse<br>Phosphorylation of CD3 and TCR zeta chains<br>PD-1 signaling<br>Generation of second messenger molecules | Immune effector function<br>T cell interaction |
| **C4_0 DC** | Formation of a pool of free 40S subunits<br>GTP hydrolysis and joining of the 60S ribosomal subunit<br>Cap-dependent Translation Initiation<br>Eukaryotic Translation Initiation<br>L13a-mediated translational silencing of Ceruloplasmin expression | Protein synthesis and translational control |

*(Continued)*

**Table 3.** (Continued)

| Myeloid Sub-cluster | Top Reactome Pathways by z-score | Putative Role(s) in Mammary/Milk Environment |
|---|---|---|
| **C4_1 DC** | ATF4 activates genes in response to endoplasmic reticulum stress<br>PERK regulates gene expression<br>Interleukin-10 signaling<br>Chemokine receptors bind chemokines<br>Senescence-Associated Secretory Phenotype (SASP) | Stress response<br>Immunoregulation<br>Chemotaxis |
| **C5_0 DC** | Eukaryotic Translation Elongation<br>Peptide chain elongation<br>Eukaryotic Translation Termination<br>Formation of a pool of free 40S subunits<br>Selenocysteine synthesis | Protein synthesis and translational control |
| **C9_0 DC** | Neutrophil degranulation<br>ROS and RNS production in phagocytes<br>Interleukin-10 signaling<br>Regulation of IFNγ signaling<br>Signaling by Interleukins | Immune effector function<br>Immunoregulation |
| **C5_1 DC** | Interleukin-10 signaling<br>Interleukin-4 and Interleukin-13 signaling<br>Neutrophil degranulation<br>Trafficking and processing of endosomal TLR<br>Metallothioneins bind metals | Immunoregulation<br>Pathogen sensing<br>Metal homeostasis |
| **C6_0 Epithelial-like cells** | Neutrophil degranulation<br>Apoptotic cleavage of cell adhesion proteins<br>Response to metal ions<br>Metallothioneins bind metals<br>Attenuation phase | Epithelial stress response<br>Redox regulation<br>Immune effector function |

C7_0, C7_1), M1 macrophages (C1_0, C1_1), monocytes (C3_1), and dendritic cells (C4_0, C5_0, C5_1). The SPP1 pathway, with a total signaling score of 63.82, average strength of 0.29, and 223 interactions, was also widely used by M2 macrophages and dendritic cells. MIF signaling followed with a total score of 59.13, average strength of 0.31, and 189 distinct interactions, and was sent by monocytes and M1 macrophages. Although the MHC-I pathway had fewer interactions [12], it displayed the highest average interaction strength at 0.55, with contributions from M1 and M2 macrophages, dendritic cells, and epithelial-like myeloid cells. The CCL chemokine axis also emerged as a key signaling route with a total score of 14.31 and an average strength of 0.27.

## Cell-cell communication patterns of the human milk myeloid compartment

To map the outgoing signaling landscape of myeloid and other immune cells in human milk, we performed cell-cell communication inference analysis using ligand-receptor interaction modeling [13]. All data is available in supplemental data 3. Note that Pattern 2 (Fig 7) consisted entirely of lymphocyte sub-clusters and is not described here.

River plot visualizing dominant patterns of outgoing cell-cell communication as inferred from single-cell transcriptomes. Cell-cell communication analysis was performed using the CellChat R package. Sub-clusters (left) were assigned to 4 dominant signaling patterns (middle), which were each linked to a set of signaling pathways (right). Pathway groupings reflect similarities in outgoing signaling behavior across clusters. Flow widths are illustrative and do not reflect interaction strength.

**Pattern 1: signals from C3_1 monocytes, C0_0/C0_1/C3_0/C7_0 M2 macrophages.** This pattern included ligands that coordinate vascular inflammation, metabolic adaptation, and macrophage plasticity, including GALECTIN, APP, ANNEXIN, THBS, GRN, LAMININ, SEMA3, RESISTIN, JAM, TGF-β, PECAM1, PECAM2, CysLTs, NRG1, TWEAK,

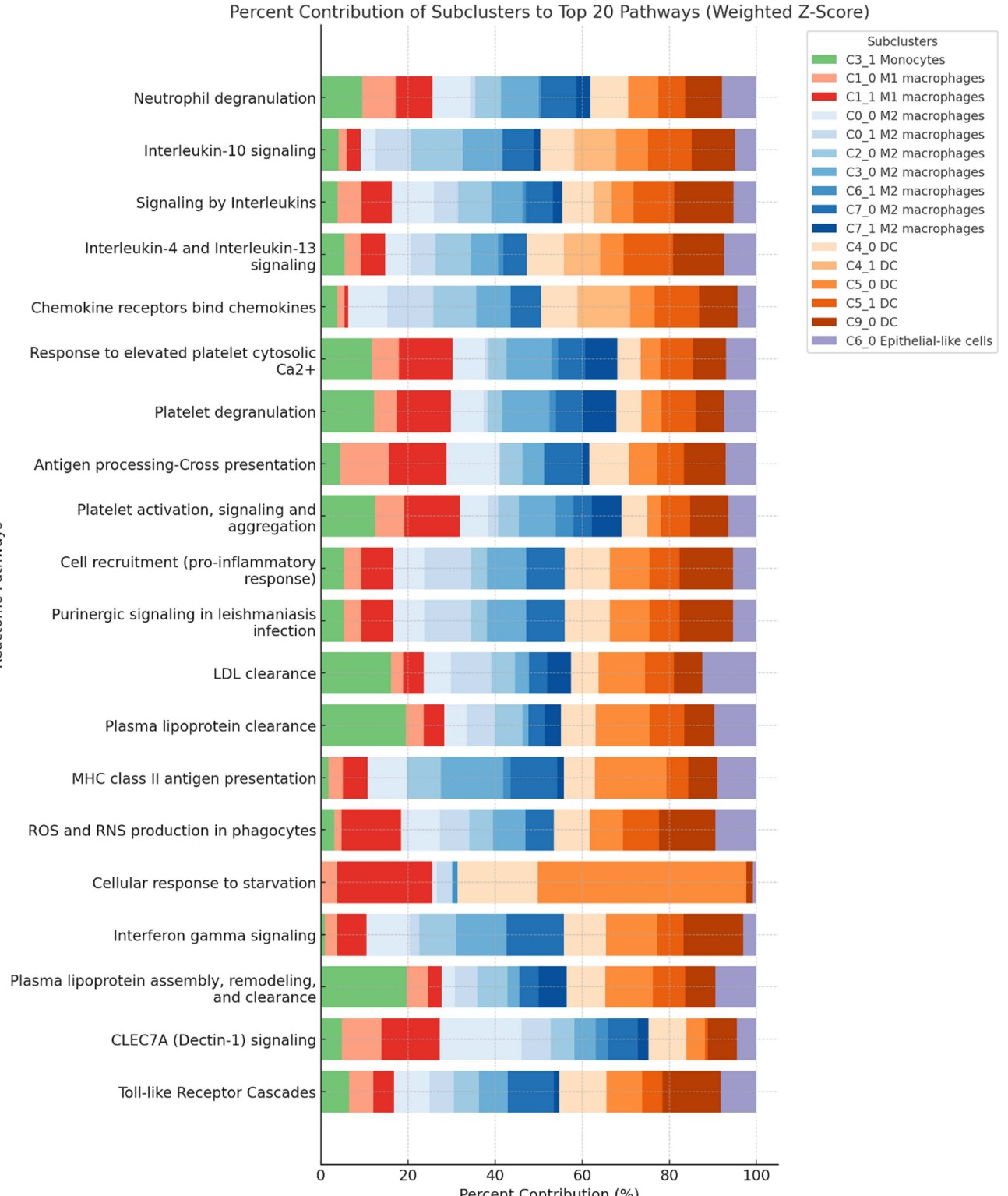

**Fig 5. Percent contributions of myeloid sub-clusters to the top 20 Reactome pathways ranked by weighted z-score.** Pathways are ordered from greatest to least weighted z-score and restricted to positive enrichment values. Weighted z-score was determined by multiplying the total additive z-score (restricted to positive contributions) by the number of contributing sub-clusters.

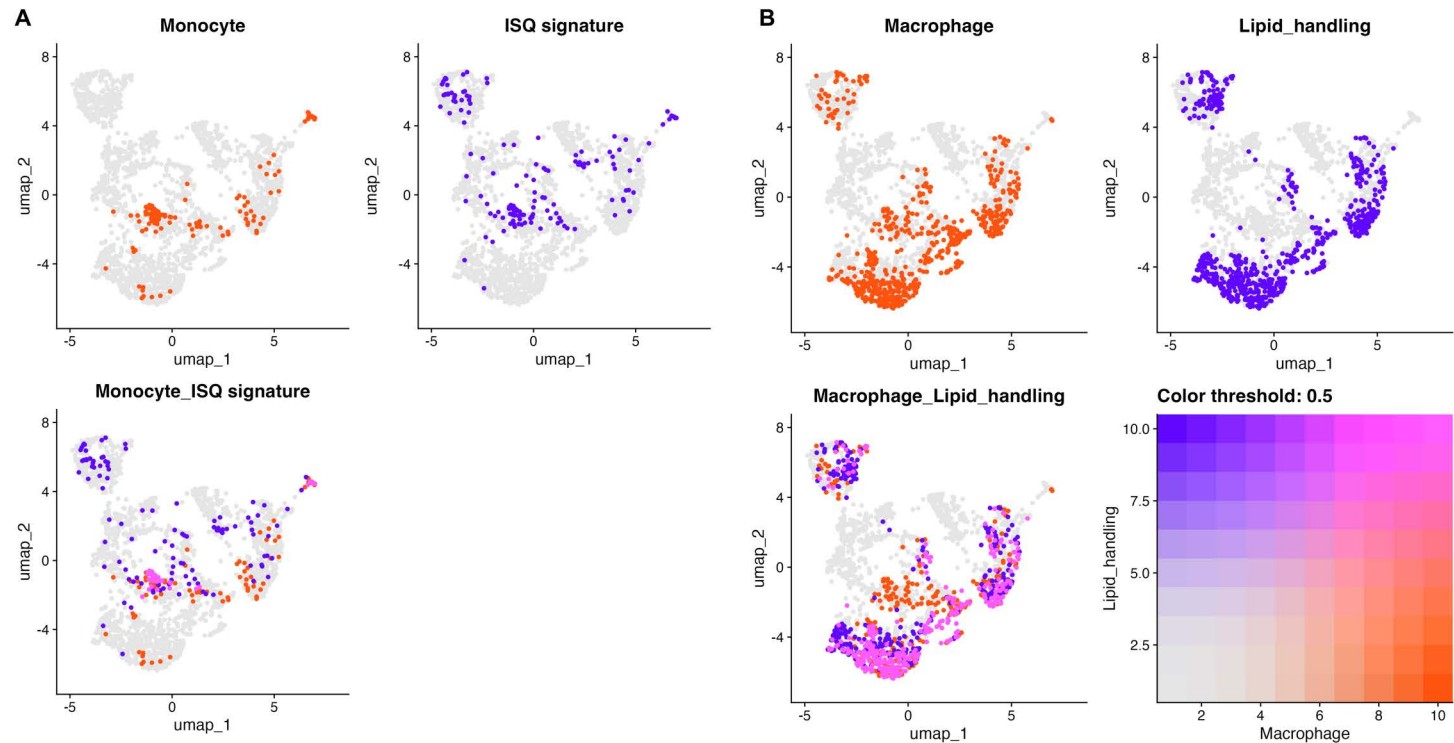

**Fig 6. Correlation analysis of canonical monocytes and macrophage markers with various lipid-handling transcriptomic signatures.** Monocyte (A; S100A9, FCGR1A, ITGAM) and macrophage (B; FCGR3A, CD68, MAFB, CD14) marker genes are shown in red, lipid signatures (INSIG1, SLC7A5, QSOX1 ("ISQ") for monocytes or APOE, LIPA, APOC1, NCEH1, SOAT1, A2M for macrophages) are shown in blue, and co-expression is shown in fuchsia. A threshold of 0.5 was applied to emphasize cells with concurrent enrichment in both dimensions, highlighting regions where identity and metabolic state overlap in the UMAP space.

SN (Siglec-1), and 12-oxo-LTB4. GALECTIN and ANNEXIN family proteins regulate cell adhesion, apoptosis, and resolution of inflammation. APP and GRN proteins modulate lysosomal function and tissue remodeling. SEMA3 proteins guide immune infiltration and adipose tissue remodeling. RESISTIN, a macrophage-derived adipokine, drives systemic inflammation and metabolic dysfunction. JAM and PECAM molecules support endothelial barrier regulation and leukocyte transmigration. THBS1 interacts with integrins and CD47 to regulate leukocyte adhesion and inflammation. LAMININ contributes to extracellular matrix remodeling and immune cell positioning. CysLTs and 12-oxo-LTB4 amplify leukotriene-mediated inflammatory signaling and phagocytosis. TGF-β orchestrates immune tolerance, fibrosis, and tissue repair via SMAD-dependent signaling. NRG1 promotes epithelial regeneration through fibroblast-macrophage crosstalk. TWEAK, acting through Fn14, modulates tissue regeneration and inflammation via NF-κB and MAPK. SN (Siglec-1) recognizes sialylated pathogens and apoptotic cells, contributing to innate immune sensing and antigen capture.

**Pattern 3: signals originating from C9_0/C4_0/C5_0/C5_1 DCs.** This signaling axis reflected interplay between pro-immune co-stimulation and immune resolution, encompassing CD80, GAS, CD137, SEMA7, CD39, CD70, OX40, PD-L1, LXA4, Calcitriol, AGRN, and desmosomal ligands. CD80 provides dual signaling to either activate (via CD28) or inhibit (via CTLA-4) T cells. GAS sequences activate STAT1-dependent transcription in response to IFN-γ, enhancing antimicrobial function and Th1 polarization. CD137 and OX40 act as co-stimulatory receptors promoting T cell proliferation and survival, while CD70 supports effector T cell differentiation. SEMA7 facilitates leukocyte migration and modulates cytokine production, while CD39 contributes to the adenosinergic checkpoint by hydrolyzing extracellular ATP. PD-L1 tempers

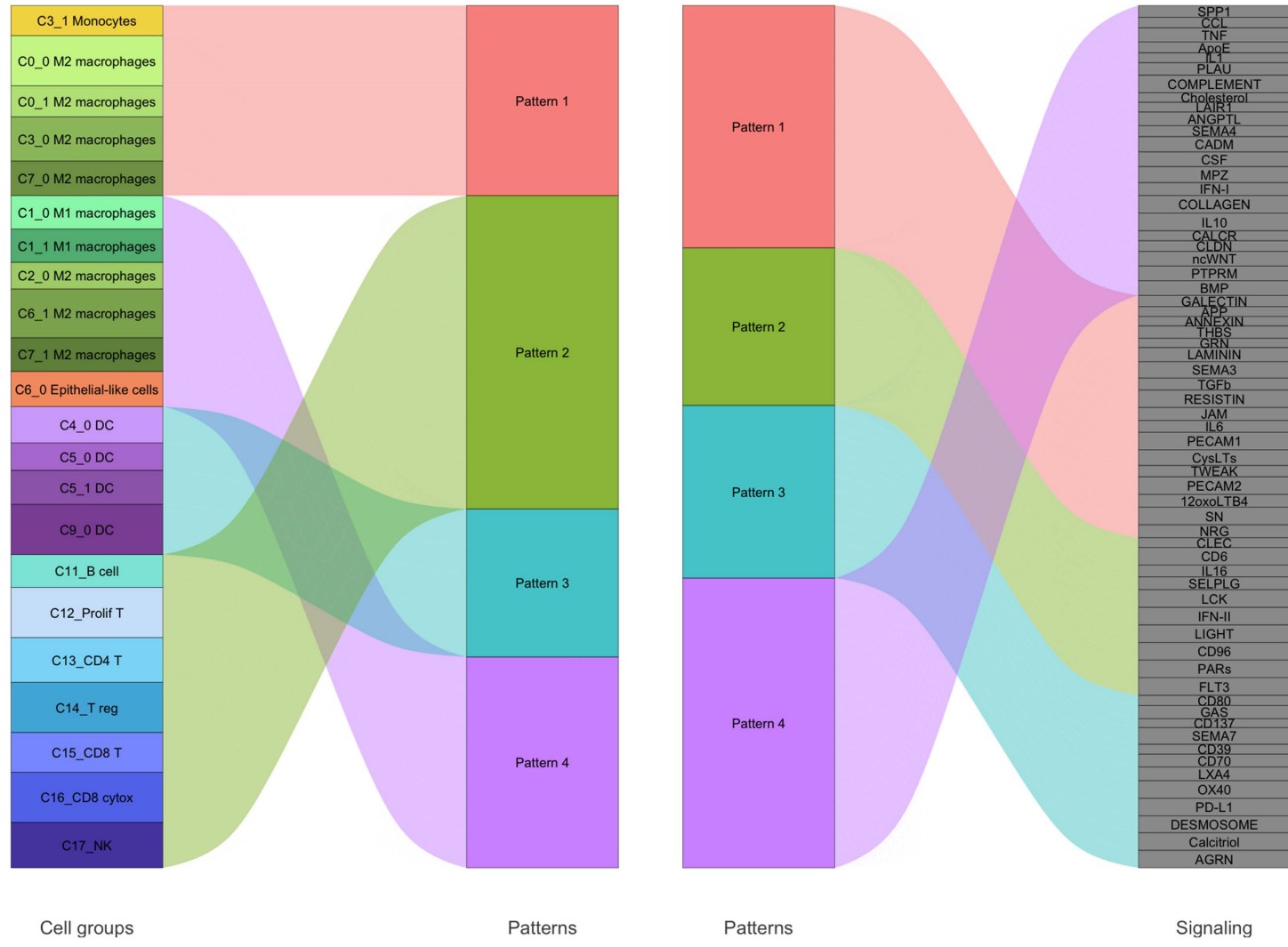

**Fig 7. Outgoing signaling patterns inferred from milk-derived leukocyte sub-clusters.**

cytotoxic immune responses by mediating T cell exhaustion. LXA4 initiates resolution of inflammation by suppressing leukocyte trafficking and promoting anti-inflammatory macrophage phenotypes. Calcitriol, the active form of vitamin D3, suppresses pro-inflammatory cytokine production and promotes immune tolerance. Structural proteins such as AGRN and desmosomal ligands maintain the epithelial-immune interface and tissue integrity during inflammation.

**Pattern 4: signals originating from C1_0/C1_1 M1 macrophages, C2_0/C6_1/C7_1 M2 macrophages, and C6_0 epithelial-like cells.** This signaling pattern was defined by a strongly inflammatory and chemotactic ligand profile, including SPP1, CCL chemokines, TNF, ApoE, IL1, PLAU, COMPLEMENT, cholesterol, LAIR1, ANGPTL, SEMA4, CADM, CSF, MPZ, IFN-I, COLLAGEN, IL10, CALCR, CLDN, ncWNT, PTPRM, and BMP. SPP1 (osteopontin) enhances fibroblast activation and macrophage-driven inflammation. CCL chemokines mediate monocyte and lymphocyte trafficking into inflamed tissues. TNF and IL1B are central drivers of inflammasome activity and inflammatory gene expression. ApoE links lipid metabolism to immune function, supporting cholesterol efflux and foam cell regulation. PLAU and COMPLEMENT components promote proteolysis, apoptotic cell clearance, and modulation of chronic inflammation.

Cholesterol metabolism provides precursors for inflammatory lipid mediators. LAIR1 acts as an inhibitory receptor dampening excessive immune activation, while ANGPTL proteins modulate angiogenesis and metabolic signaling. SEMA4 family ligands guide immune cell migration and T cell costimulation. CADM and PTPRM proteins stabilize cell-cell adhesion and intracellular signaling. CSF family cytokines regulate myeloid development and survival, while MPZ participates in adhesion processes. IFN-I induces antiviral and pro-inflammatory gene expression. COLLAGEN engages integrins to influence adhesion and matrix remodeling. IL10 acts as a key anti-inflammatory cytokine that enforces macrophage quiescence. CALCR contributes to osteoimmune regulation, while CLDN proteins maintain epithelial barrier integrity. ncWNT ligands polarize macrophages toward repair-associated phenotypes, and BMP signaling fosters stromal regeneration and immune resolution.

## Discussion

Given the intentionally small sample size of this study, which was designed as a pilot (n = 9 healthy lactating individuals from NYC), and the limited demographic diversity of the cohort, these data are not intended to capture the full spectrum of immunological variability in human milk across populations. In particular, the predominance of white participants and recruitment from a single geographic region constrain generalizability and preclude meaningful assessment of how maternal genetics, environmental exposures, or socioeconomic factors may shape milk immune composition. In addition, variability in postpartum collection time represents an important biological variable that may contribute to the observed heterogeneity in immune cell frequencies [10]. Differences among samples observed in our dataset, such as the elevated CD45+ cell frequency in a single donor, should be interpreted cautiously and not attributed to any single factor. Importantly, rather than attempting to resolve the relative contributions of various maternal characteristics or environmental influences, this study was designed as an initial high-resolution, single-cell characterization of the myeloid compartment in human milk. Within this context, our findings provide novel insight into the transcriptional diversity and potential functional states of these cells. These results therefore serve as a foundational resource and generate testable hypotheses for larger, more diverse, and longitudinal studies specifically powered to disentangle the effects of lactation stage, maternal demographics, and environmental exposures on milk immune cell composition. In this study, leukocyte enrichment prior to scRNAseq facilitated identification of 19 transcriptionally distinct mononuclear myeloid sub-clusters. Comparison to 3 prior studies in this field – Nyquist et al [10], Cansever et al [14], and Twigger et al [15]- reveals that our work substantially expands the known cellular diversity of the milk immune compartment. This study identified a diverse set of myeloid sub-clusters in human milk that both corroborate and extend beyond the macrophage populations reported by Nyquist et al. (2022). Clusters C0_0 and C0_1 exhibited a strongly M2-skewed transcriptional profile marked by expression of MRC1, CD163, and IL10, aligning closely with the GPNMB-expressing macrophage population described by Nyquist, which demonstrated high scores for anti-inflammatory and tissue maintenance gene sets. Conversely, clusters C1_0 and C1_1 were enriched for IL1B, TNF, SPP1, and CD70, and correspond to Nyquist's sub-cluster 1 – the only macrophage population in their dataset with a dominant M1 gene signature.

Cluster C3_1 expressed FCN1, CD14, and S100A9, characteristic of classical monocytes, and likely corresponds to the CD14+ monocyte population included in Nyquist's broader myeloid annotation, although they did not resolve it into a separate sub-cluster. Cluster C6_0, marked by high levels of CSN1S1 and other milk protein transcripts, showed strong overlap with Nyquist's CSN1S1+ macrophage subset, which they hypothesized to represent milk-ingesting or epithelial-associated macrophages. In contrast, multiple macrophage sub-clusters in this dataset had no clear counterparts in Nyquist. Furthermore, this study resolved 5 DC sub-clusters (C4_0, C4_1, C5_0, C5_1, and C9_0), characterized by distinct expression of CD1C, CLEC10A, CD207, and CCRL2. Nyquist annotated ITGAX+/CD33+ and CD68+/FCER1G+ clusters likely encompassing DCs, but did not further resolve these into discrete subtypes. The identification of transcriptionally distinct DC subpopulations and additional macrophage states thus expands the cellular complexity of the human milk myeloid compartment beyond prior definitions. Cansever et al. described 3 human macrophage subsets in human

milk: Mac1 (expressing TREM2, APOE, LIPA), interpreted as lipid-associated macrophages; Mac2 (CX3CR1+, Dectin-1+, CD11c+), proposed to be lactation-induced macrophages (liMacs); and Mac3, associated with stress and immunomodulatory features. Some of our M2-like macrophages—particularly C3_0 and C6_1—shared LIPA and APOE expression with Mac1.

Twigger et al. primarily focused on epithelial diversity between lactating and non-lactating breast tissue. However, they did annotate 12 immune sub-clusters from both sources, including a CD68+/FCER1G+ myeloid cluster, three ITGAX+/CD33+monocyte/neutrophil subtypes, and two CD163+/MSR1+/C1QB+ macrophage subsets. Their macrophage clusters appear to align with broadly M2-like profiles (based on CD163 and MSR1), but no analogues of our M1, lipid-inflammatory hybrids, transitional monocytes, or epithelial-like cells were reported. Additionally, dendritic cells were not annotated in their sub-clustering. As well, our data highlights the pronounced inter-individual variability across donors. The contribution of individual donors to specific sub-clusters was often highly skewed, suggesting both biological and potentially environmental or temporal factors influencing myeloid cell composition. While the Canonical Correlation Analysis (CCA)-integration aligned major cell types across all donors, subsequent high-resolution sub-clustering revealed donor-specific clustering in certain populations. This observation may indicate the 'biological floor' of integration, where donor-specific transcriptional nuances outweigh the shared sub-state signatures. Rather than over-integrating and risking the loss of true biological variation, we maintained these clusters to reflect the natural heterogeneity of the cohort at near-single-cell resolution. Given the small sample size of this pilot study, (n = 9 healthy lactating individuals from New York City), these findings certainly do not capture the potential spectrum of immunological variability across populations differing by geography, ethnicity, age, lactation stage, health status, and other factors.

Beyond classical immune functions, several of the clusters identified in this study exhibited transcriptional programs suggestive of adaptation to the milk environment. In particular, the reactome enrichment profile for the monocyte cluster revealed transcriptional signatures centered on LDL clearance, and plasma lipoprotein assembly, remodeling, and clearance. These pathways, alongside expression of genes such as INSIG1 (a key regulator of cholesterol biosynthesis feedback) and QSOX1 (involved in disulfide bond formation and extracellular redox activity), suggest an adaptation of these cells to the lipid-rich environment of milk [16]. While not directly part of canonical lipid metabolism, QSOX1-mediated redox processes may support the structural maturation of secreted lipoproteins and stabilize extracellular proteins involved in lipid transport [17]. Additionally, ECM remodeling activity may facilitate monocyte access to lipid pools or lipoprotein complexes secreted into the alveolar lumen, particularly in the dynamic stromal context of lactation [18]. Together, these features point to a functional specialization of these monocytes in lipid sensing and clearance, potentially contributing to immune-metabolic regulation within the human milk immune compartment. As well, several other myeloid subsets suggested atypical gene expression and signaling patterns suggesting adaptation to the milk/mammary environment. Multiple macrophage sub-clusters exhibited DEGs not previously associated with myeloid cells, including NPTX2, AGRP, and CELSR1, which are typically restricted to neuronal or epithelial lineages, suggesting transcriptional imprinting by local mammary or neuroepithelial cues. ECM-modifying enzymes such as DSE, MFAP5, and HS3ST2 were broadly expressed in both M1- and M2-like macrophage subsets, potentially supporting stromal interaction or ductal migration. Metabolic reprogramming was reflected in the expression of genes like ACOX2, JAKMIP2, and SLC38A6, which mediate alternative nutrient handling and redox adaptation, consistent with the high metabolic demands of milk-resident immune cells. Several dendritic cell subsets expressed unexpected endocrine and epithelial transcripts such as ENHO, TRH, and UPK3A, again supporting the idea of local transcriptional rewiring. Reactome enrichment across sub-clusters further revealed robust upregulation of translational and surveillance programs, with high activity in ribosomal assembly, IL-10 signaling, chemokine responsiveness, and granule exocytosis in macrophage and dendritic cell subsets. These features together suggest that the human milk immune compartment is shaped by a combination of epithelial crosstalk, metabolic stress, ductal structure, and inflammatory cues, producing transcriptional states not captured in conventional blood or tissue-derived myeloid references. Such adaptations reflect a distinct immunological niche optimized for tissue

surveillance, immune modulation, and epithelial interaction in the lactating mammary gland. Overall, the weighted z-score analysis of the Reactome data suggested that the top-ranked pathways in the myeloid compartment represent two major functional axes. The first axis is immune signaling, encompassing neutrophil degranulation, cytokine and chemokine receptor pathways, interferon responses, and BCR downstream events. These were consistently driven by M2 macrophages and DCs, with selective involvement of M1 macrophages in inflammatory contexts. The second axis is biosynthetic upregulation, including ribosomal subunit assembly, translation initiation and elongation, and cotranslational targeting. These pathways were again dominated by M2 macrophages and DCs, with additional input from M1 macrophages in translational regulation. Together, these patterns highlight the coordinated engagement of macrophage and DC populations in both effector signaling and the biosynthetic machinery needed to sustain activation.

Importantly, the present analysis provides a systems-level view of intercellular communication dynamics. MHC-II signaling was the most robust pathway observed, both in terms of total volume and interaction breadth, and was active in nearly all myeloid subtypes. MHC-II molecules are central to the presentation of extracellular antigenic peptides to CD4$^+$ T helper cells, a process that orchestrates adaptive immune responses and contributes to mucosal tolerance [19]. The finding that M2 macrophages and dendritic cells were primary senders of MHC-II signals suggests that these cell types in milk may play crucial roles in antigen sampling and immune education during lactation. SPP1 (osteopontin) signaling also featured prominently and was both produced and received by macrophages and dendritic cells. Osteopontin is a multifunctional cytokine known for its roles in cell adhesion, migration, and inflammatory signaling [20]. Its presence in human milk has been associated with mucosal immune enhancement and epithelial-matrix remodeling, and the identification of this pathway in both directional axes implies that osteopontin-mediated feedback may contribute to shaping immune architecture within the mammary gland and possibly in the neonatal gut. MIF was another major axis of interaction, with widespread signaling directed at M2 macrophages and dendritic cells. MIF is a pleiotropic cytokine involved in immune cell recruitment, activation, and survival, and it plays essential roles in resolving or sustaining inflammation depending on context [21]. Its engagement with regulatory myeloid populations in milk suggests potential roles in maintaining tissue quiescence, immune tone, or adapting to microbial challenges encountered during lactation.

Although donors were healthy and samples lacked overt signs of infection or inflammation, several myeloid subclusters expressed pro-inflammatory mediators such as IL1B, TNF, and SPP1, reflecting partial activation states commonly associated with M1-like polarization. These inflammatory features have also been observed in prior studies of milk-resident immune cells, though their presence was not attributed to pathological inflammation. In the context of this dataset, such expression may instead reflect adaptation to the lipid-rich environment of milk and the ongoing structural remodeling of the lactating mammary gland. Continuous epithelial shedding, ductal turnover, and exposure to secreted lipids may provide metabolic and mechanical cues that engage macrophage sensing and surveillance pathways. The presence of inflammatory mediators in this setting likely supports roles in tissue remodeling, lipid clearance, and barrier maintenance, representing functional activation rather than a classical immune response.

A growing body of evidence indicates that maternal leukocytes transferred via milk can survive passage through the infant gastrointestinal tract and exert immunological effects beyond local mucosal surfaces. In murine models, milk-derived macrophages and T cells have been shown to migrate to secondary lymphoid organs such as the spleen, liver, and mesenteric lymph nodes, where they remain viable and functionally active [22,23]. Human data support this possibility as well, in that leukocyte concentrations in milk increase during maternal or infant infection and that these cells can express activation markers, suggesting systemic signaling or trafficking potential [24]. Moreover, milk-derived immune cells may contribute not only to mucosal protection but also to systemic immune development, particularly in the context of neonatal immune immaturity [1]. Together, these studies raise the possibility that maternal myeloid cells transferred through milk may integrate into the infant's immune system under physiologically relevant conditions. In this context, our transcriptional analysis of milk-resident myeloid cells adds mechanistic insight into the potential impact of these cells post-transfer. The presence of transcriptionally active MHC-II$^+$ antigen-presenting cells, interferon-responsive

macrophages, and regulatory dendritic cell subtypes suggests that many of these cells possess the molecular hallmarks of immune education, rather than simply passive surveillance. If such cells retain viability and functionality following ingestion, as supported by prior in vivo and in vitro work, they may serve as maternal agents of immune modulation in the infant — potentially influencing T cell differentiation, tolerance, or early-life immune memory. These implications underscore the importance of characterizing the human milk immune compartment with high resolution, and motivate future studies to directly track milk-derived myeloid cell fates and functions in the infant.

Collectively, the present data underscore the transcriptional plasticity and environmental responsiveness of the human milk myeloid compartment. The discovery of atypical gene programs, immunoregulatory signaling axes, and lipid-adapted phenotypes suggests that milk-resident myeloid cells are not merely passively transported immune cells, but actively shaped by the lactational niche. Their potential to persist beyond the infant gut and participate in immune programming further reinforces the concept that the maternal immune system extends its influence postnatally. As this study is based on single-cell transcriptomic profiling, all functional interpretations are necessarily inferred from gene expression patterns rather than directly measured at the protein or cellular behavior level. Accordingly, predicted signaling interactions, activation states, and niche-adaptive features should be interpreted as hypothesis-generating rather than definitive evidence of function. Important questions remain regarding the ontogeny, dynamics, and functional roles of these cells – particularly their responses to infection, inflammation, hormonal modulation, and environmental exposures. Additionally, it remains to be determined how these immune states influence infant mucosal immunity and long-term health outcomes. Future studies will be essential to ultimately inform evidence-based lactation strategies and the design of therapeutics benefitting the lactating population and their infants.

## Materials and methods

### Participant information

Participants were recruited for this study between 3/1/2023–10/3/2023 using social media advertisement, and informed written consent was obtained in accordance with the ethical and institutional board approval under the guidance and authorization of Mount Sinai's Program for the Protection of Human Subjects (PPHS) using an IRB-approved protocol for obtaining human milk samples (STUDY-17–01089). Individuals were eligible to have their milk samples included in this analysis if they were 18+, lactating, feeling well, and living in New York City. Participants were excluded if they had any acute or chronic health conditions affecting the immune system. Milk was expressed using manual or electric pumps in participants' homes just prior to sample pickup using clean containers. Milk was used for analysis within 12 hours of expression.

### Human milk cell purification and sorting

Fresh milk was poured into 50 mL centrifuge tubes and centrifuged at 800 × g for 10 minutes at room temperature. Following centrifugation, the fat layer was skimmed off, and the milk supernatant was carefully transferred into new, sterile tubes for future use. The cell pellets were resuspended in phosphate-buffered saline (PBS), combined into a single tube, and centrifuged at 400 × g for 5 minutes at room temperature. After discarding the supernatant, the cells were resuspended in 10 mL PBS and centrifuged again under the same conditions to ensure thorough washing. The resulting cell pellet was resuspended in PBS and total cell numbers were determined. Approximately $0.5 \times 10^6$ cells were aliquoted into a separate tube as unstained control cells and kept on ice. The remaining cells were stained with Aqua Live/Dead viability dye (Fisher; 1 μL per $10^6$ cells) and incubated for 30 minutes at 4°C in the dark. Cells were then washed with 1% bovine serum albumin (BSA) at 300 × g for 5 minutes.

For surface marker staining, cells were resuspended in 1–5 mL of anti-CD45 antibody solution (1 μL per $10^6$ cells in 1% BSA) and incubated at 4°C for 30 minutes. Following incubation, cells were washed. Nuclear staining was performed by

resuspending cells in a DRAQ5 solution (Fisher; 2.5 µL per 2.5 mL PBS per $10^6$ cells) and incubating for 30 minutes at room temperature. A final wash step was carried out by centrifugation at $300 \times g$ for 5 minutes, followed by resuspension in PBS. Prior to analysis all samples were filtered into FACS sorting tubes equipped with 40 µm mesh filters and kept on ice until sorting. Cell suspension was placed on ice and resuspended using a wide-bore tip, with a P200 pipette set to 100 µL, as the total volume of the sample was approximately 150 µL. Following resuspension, the sample was quality controlled (QC) by mixing 5 µL of trypan blue dye with 5 µL of the resuspended cell suspension in a 1:1 ratio (total volume = 10 µL). The mixture was then loaded onto a C-chip hemocytometer and visualized under an EVOS M7000 microscope, set at 10x magnification in transmitted light mode. Cell viability was determined by counting the number of live and dead cells per microliter of suspension. The live cell count was expressed as a ratio of live cells to the total number of cells. To improve single cell suspension quality, it was filtered through a FlowMi filter to break up clumps and remove debris.

## Sorted sample preparation

For sorted samples with low cell counts, cells were transferred to a 1.5 mL Eppendorf tube and pelleted by centrifugation at $300 \times g$ for 5 minutes. The supernatant was retained on ice, and the pellet was resuspended in 50 µL of PBS containing 0.04% BSA. The resuspended sample was re-QC'd to ensure the cell count reached the required minimum loading concentration. Nearly all sorted samples were re-concentrated for optimal results.

## Single cell encapsulation sequencing

scRNAseq was performed on these samples using the Chromium platform (10x Genomics, Pleasanton, CA) with the 3' gene expression (3' GEX) V3 kit. The cell concentration (cells/µL) determined the loading concentration, based on a target recovery of 6,000 cells, as outlined in the 10x Genomics Chromium Single Cell 3' Reagent User Guide (CG000315). Following successful gel bead in emulsion (GEM) formation and barcoding (with unique molecular identifier, UMI, encapsulation), the sample proceeded to cDNA amplification and gene expression library construction according to manufacturer's instructions. Libraries were quantified using Bioanalyzer (Agilent) and QuBit (Thermofisher) analysis and were sequenced in 2x100 paired end mode on a NovaSeq instrument (Illumina, San Diego, CA) targeting a depth of at least 50,000 reads per cell. Sequencing data was aligned and quantified using the Cell Ranger Single-Cell Software Suite (version 7.1, 10x Genomics) against the provided GRCh38 reference genome.

## scRNA-seq data analysis

The R-based package Seurat (v3.1.1) was used to process single-cell RNA sequencing data [25]. Quality control (QC) metrics were applied to evaluate the relationships between read counts, transcript counts, and mitochondrial gene ratios. Read and transcript counts were expected to be roughly correlated, and mitochondrial gene expression was limited to below 25%. Cells identified as outliers, including those expressing < 2000 or > 100000 total molecules, or < 200 unique transcripts, or with a log10(transcript)/log10(UMI) ratio below 0.8, were excluded from further analysis. Doublets were detected and removed using scDblFinder [26]. Genes detected in < 0.5% of cells were removed.

The *SCTransform* function in Seurat was utilized to perform normalization, scaling, and variance stabilization of the data. Subsequently, multiple integration approaches were applied to the datasets to address potential sample, donor, or batch effects. Known cell-type-specific markers and metadata were visualized to assess the suitability of each integration method. As a result, Canonical Correlation Analysis (CCA) integration was chosen, which identifies correlated structures across the datasets to align them in a common space. The so integrated data object was used for all subsequent analyses, including differential expression, marker identification, and other downstream assays. Dimensionality reduction was performed using principal component analysis (PCA) of the most variable 2000 genes. The first 30 principal components and were used for uniform manifold approximation and projection (UMAP) and clustering (resolution 0.5). Cell types were assigned

by comparing 1) manual annotation using established canonical markers, 2) automated machine learning using annotated published data [10,27,28] to train a random forest classifier, and 3) the Unicell deconvolution software, which uses a deep learning model pre-trained on a fully integrated scRNA-seq training database, comprising more than 840 cell types from 899 studies [29]. Differential expression analysis was performed using the *FindMarkers* function (negative binomial generalized linear model, correcting for donor effect) in Seurat. Gene set over-representation analysis was performed using the *enricher* function in the clusterProfiler R package, in conjunction with the Reactome and msigdbr Hallmark databases [13,30,31]. Differentially expressed genes (DEGs) were queried against all accessible peer-reviewed literature as well as the Gene Expression Omnibus (GEO) and the Human Protein Atlas (HPA), using relevant search terms. Functional studies and transcriptomic datasets, including single-cell and bulk RNA-seq publications, were screened. DEGs were classified as atypical/non-canonical for a given cell type if they lacked prior characterization in any relevant lineage. This process allowed identification of DEGs likely to reflect context-specific, tissue-adapted, or novel immune functions. Additionally, to explore pathways beyond individual sub-clusters, we derived a weighted z-score by multiplying the sum of positive z-scores (as output by clusterProfiler) by the number of contributing sub-clusters. This allowed us to highlight pathways that were not only strongly activated but also broadly represented across multiple myeloid cell states.

Cell-cell interaction analysis was performed using the CellChat R package, which infers cell-cell communication through the expression of known receptors and ligands, assigns each interaction with a probability value and performs permutation tests [12]. Interaction strength was quantified using three complementary metrics: the total signaling score, defined as the sum of probabilities across all sender-receiver pairs for a given pathway; the average interaction strength, calculated as the mean probability across all interactions within that pathway; and the number of unique interactions per pathway. Functional descriptions of signaling pathways and ligand families were derived from Reactome [13].

## Supporting information

**S1 Data. Myeloid Cell type DEGs.**
(XLSX)

**S2 Data. Myeloid Subcluster DEGs.**
(XLSX)

**S3 Data. Myeloid Subcluster Reactome.**
(XLSX)

**S4 Data. Cellchat data.**
(XLSX)

## Acknowledgments

We are grateful to our study participants for providing their invaluable milk samples for this study.

## Author contributions

**Conceptualization:** Kristin Beaumont, Rebecca L. Powell.

**Data curation:** Kristin Beaumont, Rebecca L. Powell.

**Formal analysis:** Nadine Schrode, Kristin Beaumont, Rebecca L. Powell.

**Funding acquisition:** Rebecca L. Powell.

**Investigation:** Nadine Schrode, Xiaoqi Yang, Alisa Fox, Disha Chowhan, Claire DeCarlo, Kristin Beaumont, Rebecca L. Powell.

**Methodology:** Nadine Schrode, Xiaoqi Yang, Alisa Fox, Disha Chowhan, Claire DeCarlo, Kristin Beaumont, Rebecca L. Powell.

**Writing – original draft:** Rebecca L. Powell.

**Writing – review & editing:** Nadine Schrode, Xiaoqi Yang, Kristin Beaumont, Rebecca L. Powell.

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
