## [Decision Letter · Decision Letter 0]

19 Jan 2026

PONE-D-25-61210Single-cell transcriptomics of leukocyte-enriched human milk reveals a highly diverse and adapted myeloid compartmentPLOS One

Dear Dr. Powell,

Thank you for submitting your manuscript to PLOS ONE. After careful consideration, we feel that the manuscript has merit; however, it does not fully meet PLOS ONE’s publication criteria in its current form. Therefore, we request that you consider the reviewers’ comments and address them to the extent possible. We invite you to submit a revised version of the manuscript that addresses the points raised during the review process.

We look forward to receiving your revised manuscript.

Kind regards,

Srinivasa Reddy Bonam

Academic Editor

PLOS One

“Campbell Foundation grant.”

“We are grateful to our study participants for providing their invaluable milk samples for this study. This study was funded by a grant from the Campbell Foundation.”

“Campbell Foundation grant.”

Reviewers' comments:

Reviewer's Responses to Questions

**Comments to the Author**

1. Is the manuscript technically sound, and do the data support the conclusions?

Reviewer #1: Yes

Reviewer #2: Yes

2. Has the statistical analysis been performed appropriately and rigorously? 

Reviewer #1: Yes

Reviewer #2: Yes

3. Have the authors made all data underlying the findings in their manuscript fully available?

Reviewer #1: Yes

Reviewer #2: Yes

4. Is the manuscript presented in an intelligible fashion and written in standard English?

Reviewer #1: Yes

Reviewer #2: Yes

5. Review Comments to the Author

Reviewer #1: Dr. Powell and team present a single-cell transcriptomic study investigating the diversity and tissue-specific adaptation of immune cells in human breast milk, with a strong focus on mononuclear myeloid subsets. By analyzing over 23,000 CD45-enriched cells from nine healthy lactating individuals using scRNA-seq, they reveal a highly diverse myeloid population. The study identifies several macrophage and dendritic cell sub-clusters, notes donor-specific subpopulation patterns, and highlights gene expression profiles linked to lipid processing and tissue adaptation.

The novelty of the current study is positioned as an immune-focused complement to earlier milk scRNA-seq studies, which lacked immune cell enrichment. It offers a more detailed view of the myeloid compartment in human milk and includes interpretations of cell communication networks and niche-related programs.

The experimental design follows standard scRNA-seq practices. The sample preparation process, covering fresh milk collection, immune cell enrichment, and exclusion of unwanted cells, is well-documented, suggesting strong experimental rigor. Sequencing was performed using 10x Genomics and NovaSeq, with a target of at least 50,000 reads per cell.

The computational analysis includes Cell Ranger for alignment and Seurat for quality control, normalization (via SCTransform), donor/batch integration (using CCA), dimensionality reduction (PCA/UMAP), clustering (Louvain, resolution 0.5), and a multi-step cell annotation process. This includes canonical marker analysis, machine learning (random forest classifier), and deep learning (Unicell deconvolution). Differential expression was handled using Seurat’s Wilcoxon test and pathway enrichment via clusterProfiler, with Reactome and Hallmark databases. CellChat was used to infer ligand-receptor interactions, with clear scoring and permutation testing.

Despite the study’s strengths, several areas could improve their technical depth and interpretability of the seminal findings:

1. The sample size (9 donors) is small and lacks broad diversity, being predominantly white and from a single location (New York City). This restricts the generalizability of findings. Additionally, there's significant variation in postpartum collection times, which might explain some observed differences; for example, the only Asian donor had significantly higher CD45+ cells (27%) compared to others (<9%). This raises questions about whether postpartum timing, milk collection, or maternal genetics is what majorly affects immune cell compositional diversity in milk. While the Good et al study cited in reference 10 of the manuscript alludes to this, the authors can revise their text so that the clarity and novelty of these new findings are well articulated.

2. In Section 2.3.1, the authors mention clusters dominated by single donors (e.g., C1_2 from S207; C2_0 and C6_1 from S205). Similar patterns are noted in epithelial-like clusters. This raises statistical concerns around pseudo-replication if donor identity is not properly modeled. Though the authors mention CCA integration, they do not clearly describe donor-aware differential expression methods (e.g., mixed models or pseudo-bulk approaches). More details are needed here to enhance the clarity of this approach.

3. While the manuscript attempts to biologically validate clusters expressing both epithelial and immune markers, donor-restricted patterns (e.g., C6_2 and C6_3 from single donors) suggest the possibility of technical artifacts like doublets or ambient RNA. The authors should clarify what supports these clusters being real biological states rather than artifacts.

4. As a follow-up to question 3 above, the authors state that debris, doublets, and dead cells were excluded during sorting. There is, however, no mention of using computational methods to detect doublets or decontaminate ambient RNA in the analysis pipeline. This omission should be addressed to help readers understand observed vs expected doublet rates.

5. There are numerical issues and inconsistencies in sample labels. For example, the manuscript mentions 23,443 post-QC cells, but Table 1 lists 30,704 for sample S205 alone. Also, sample naming varies between upper and lowercase (S205 vs s206) and includes mismatches (S204 vs S204C). These should be corrected for clarity.

6. The manuscript frequently describes cells as ‘atypical’ or ‘non-canonical’ without clear classification criteria. It would be helpful for the authors to define what qualifies as atypical (e.g., by listing specific markers, inclusion/exclusion rules, and supporting evidence from the literature).

7. While the study does a good job of inferring potential signaling pathways and adaptation states, these remain transcriptomic predictions. Strong conclusions about functional roles (e.g., niche adaptation or signaling behaviors) should be tempered unless validated by orthogonal methods such as flow cytometry, protein assays, or imaging. Currently, the best the authors can claim is that the data support hypotheses, not that they conclusively prove them.

Minor aspect: The weighted z-score method for Reactome pathway analysis needs a clearer explanation, including how the scores were derived and whether corrections for multiple testing and pathway size were applied.

Ethics & Integrity: The study appears ethically sound. It includes informed consent, IRB approval, and funding disclosures. No integrity concerns were noted.

In all, this is a technically sound and potentially valuable study offering a high-resolution map of myeloid diversity in breast milk, with important findings on donor variability and tissue-specific transcriptional programs. However, improvements are needed in areas like donor-aware statistical modeling, handling of potential doublets/artifacts, and clearer separation between inference and confirmed function, with some interpretations in the text.

Reviewer #2: This manuscript presents a novel and comprehensive single-cell transcriptomic analysis of myeloid cells in leukocyte-enriched human milk, addressing a critical gap in understanding the diversity and functional adaptation of these immune cells. The identification of 19 transcriptionally distinct mononuclear myeloid subpopulations, including specialized macrophage, dendritic cell, monocyte, and epithelial-like clusters, significantly expands prior knowledge of the milk immune compartment. Particularly compelling is the demonstration of lipid-rich environment adaptation (e.g., LDL clearance pathways, APOE/INSIG1 expression) and extensive intercellular communication (MHC-II, SPP1, MIF signaling), which provide valuable insights into how lactational niches shape immune cell function. The donor-specific heterogeneity observed also highlights personalized immune composition during lactation, a noteworthy finding with implications for infant health and donor milk selection.

The study design is rigorous, with leukocyte enrichment overcoming limitations of previous scRNAseq studies, and the combination of clustering, Reactome enrichment, and CellChat analysis offers robust mechanistic support. Minor suggestions include expanding on the functional relevance of atypical gene expression (e.g., neuronal/epithelial transcripts in myeloid cells) and discussing how donor demographics (e.g., limited ethnic diversity, small sample size) may influence generalizability. Overall, this work advances our understanding of human milk’s immunological complexity and provides a high-resolution framework for future studies. It is well-suited for publication in PLOS ONE, aligning with the journal’s focus on broad, impactful research in life sciences.

6. PLOS authors have the option to publish the peer review history of their article (what does this mean?). If published, this will include your full peer review and any attached files.

Reviewer #1: No

Reviewer #2: No

---

## [Author Response · Author response to Decision Letter 1]

31 Mar 2026

Response to Review

1. The sample size (9 donors) is small and lacks broad diversity, being predominantly white and from a single location (New York City). This restricts the generalizability of findings. Additionally, there's significant variation in postpartum collection times, which might explain some observed differences; for example, the only Asian donor had significantly higher CD45+ cells (27%) compared to others (<9%). This raises questions about whether postpartum timing, milk collection, or maternal genetics is what majorly affects immune cell compositional diversity in milk. While the Good et al study cited in reference 10 of the manuscript alludes to this, the authors can revise their text so that the clarity and novelty of these new findings are well articulated.

We appreciate this highly valid critique and fully agree. This concern is now more clearly discussed in the manuscript (p. 26). Given the intentionally small sample size of this study, which was designed as a pilot (n=9 healthy lactating individuals from NYC), and the limited demographic diversity of the cohort, these data are not intended to capture the full spectrum of immunological variability in human milk across populations. In particular, the predominance of white participants and recruitment from a single geographic region constrain generalizability and preclude meaningful assessment of how maternal genetics, environmental exposures, or socioeconomic factors may shape milk immune composition. In addition, variability in postpartum collection time represents an important biological variable that may contribute to the observed heterogeneity in immune cell frequencies. Differences among samples observed in our dataset, such as the elevated CD45⁺ cell frequency in a single donor, should be interpreted cautiously and not attributed to any single factor. Importantly, rather than attempting to resolve the relative contributions of various maternal characteristics or environmental influences, this study was designed as an initial high-resolution, single-cell characterization of the myeloid compartment in human milk. Within this context, our findings provide novel insight into the transcriptional diversity and potential functional states of these cells. These results therefore serve as a foundational resource and generate testable hypotheses for larger, more diverse, and longitudinal studies specifically powered to disentangle the effects of lactation stage, maternal demographics, and environmental exposures on milk immune cell composition.

2. In Section 2.3.1, the authors mention clusters dominated by single donors (e.g., C1_2 from S207; C2_0 and C6_1 from S205). Similar patterns are noted in epithelial-like clusters. This raises statistical concerns around pseudo-replication if donor identity is not properly modeled. Though the authors mention CCA integration, they do not clearly describe donor-aware differential expression methods (e.g., mixed models or pseudo-bulk approaches). More details are needed here to enhance the clarity of this approach.

The reviewer raises an important point. To mitigate potential donor effects, the differential expression analysis was performed using a negative binomial generalized linear model, adding donor as latent variable. This is now clarified in the Methods (p. 39).

3. While the manuscript attempts to biologically validate clusters expressing both epithelial and immune markers, donor-restricted patterns (e.g., C6_2 and C6_3 from single donors) suggest the possibility of technical artifacts like doublets or ambient RNA. The authors should clarify what supports these clusters being real biological states rather than artifacts.

We appreciate this comment, and have now clarified in the text (p. 30 and 39) that while the Canonical Correlation Analysis (CCA)-integration aligned major cell types across all donors, subsequent high-resolution sub-clustering revealed donor-specific clustering in certain populations. This observation may indicate the 'biological floor' of integration, where donor-specific transcriptional nuances outweigh the shared sub-state signatures. Rather than over-integrating and risking the loss of true biological variation, we maintained these clusters to reflect the natural heterogeneity of the cohort at near-single-cell resolution.

4. As a follow-up to question 3 above, the authors state that debris, doublets, and dead cells were excluded during sorting. There is, however, no mention of using computational methods to detect doublets or decontaminate ambient RNA in the analysis pipeline. This omission should be addressed to help readers understand observed vs expected doublet rates.

To confirm previous QC results, we employed an additional doublet detection package (p. 38). scDblFinder did indeed identify additional doublets and we consequently removed subcluster 6_3. This has now been updated in the text and figures.

5. There are numerical issues and inconsistencies in sample labels. For example, the manuscript mentions 23,443 post-QC cells, but Table 1 lists 30,704 for sample S205 alone. Also, sample naming varies between upper and lowercase (S205 vs s206) and includes mismatches (S204 vs S204C). These should be corrected for clarity.

Thank you for catching these inconsistencies. We have now clarified in Table 1 the cell counts after FACS and filtering and added a column for the cell counts after QC, which was not included previously. These numbers add up to the 23,443 mentioned. As well, uppercase sample IDs have been made consistent where applicable.

6. The manuscript frequently describes cells as ‘atypical’ or ‘non-canonical’ without clear classification criteria. It would be helpful for the authors to define what qualifies as atypical (e.g., by listing specific markers, inclusion/exclusion rules, and supporting evidence from the literature).

We thank the reviewer for highlighting the need for greater clarity in how atypical/non-canonical gene expression was defined. We have revised the Methods and Results sections to highlight that differentially expressed genes were evaluated against curated reference datasets, including the Gene Expression Omnibus and Human Protein Atlas, as well as peer-reviewed literature (p. 7 and 39). Genes were considered non-canonical/atypical if they showed minimal prior association with myeloid lineages, were predominantly linked to non-myeloid cell types, or represented context-restricted expression not previously described in immune cells.

7. While the study does a good job of inferring potential signaling pathways and adaptation states, these remain transcriptomic predictions. Strong conclusions about functional roles (e.g., niche adaptation or signaling behaviors) should be tempered unless validated by orthogonal methods such as flow cytometry, protein assays, or imaging. Currently, the best the authors can claim is that the data support hypotheses, not that they conclusively prove them.

We thank the reviewer for this important point and agree that our functional interpretations are based on transcriptomic inference rather than direct experimental validation. In response, we have revised the Discussion to more explicitly clarify this (p. 35). In addition, we have systematically revised the manuscript to temper causal language (e.g., replacing “demonstrate” with “suggest”) to better reflect the inferential nature of the data.

8. Minor aspect: The weighted z-score method for Reactome pathway analysis needs a clearer explanation, including how the scores were derived and whether corrections for multiple testing and pathway size were applied.

We apologize for the confusion. As now clarified in Methods (p. 39-40), over-representation analysis of significant DEGs in Reactome pathways was performed using the clusterProfiler package to determine corrected statistical significance (FDR) and the directionality of that significance (z-score). To explore pathways beyond individual sub-clusters, we derived a weighted z-score by multiplying the sum of positive z-scores by the number of contributing sub-clusters. This allowed us to highlight pathways that were not only strongly activated but also broadly represented across multiple myeloid cell states.

---

## [Decision Letter · Decision Letter 1]

22 Apr 2026

Single-cell transcriptomics of leukocyte-enriched human milk reveals a highly diverse and adapted myeloid compartment

PONE-D-25-61210R1

Dear Dr. Rebecca L. Powell,

We’re pleased to inform you that your manuscript has been judged scientifically suitable for publication and will be formally accepted for publication once it meets all outstanding technical requirements.

Kind regards,

Srinivasa Reddy Bonam

Academic Editor

PLOS One

---

## [Editor Report · Acceptance letter]

PONE-D-25-61210R1

PLOS One

Dear Dr. Powell,

I'm pleased to inform you that your manuscript has been deemed suitable for publication in PLOS One. Congratulations! Your manuscript is now being handed over to our production team.

Kind regards,

on behalf of

Dr. Srinivasa Reddy Bonam

Academic Editor

PLOS One